# The Gap in Sustainable Food Services in Public Institutions: The Perceptions of Young Consumers from Public Universities in the Madrid Region (Spain)

**DOI:** 10.3390/foods12224103

**Published:** 2023-11-12

**Authors:** José Luis Cruz Maceín, Mohamed Amine Hocine, Verónica Hernández-Jiménez, José Pablo Zamorano Rodríguez, Samir Sayadi Gmada

**Affiliations:** 1Instituto Madrileño de Investigación y Desarrollo Rural, Agrario y Alimentario (IMIDRA) Comunidad de Madrid, Autovía A2 Km 38,200, Alcalá de Henares, 28800 Madrid, Spain; amiinehcn@gmail.com (M.A.H.); pablo.zamorano.rodriguez@madrid.org (J.P.Z.R.); 2International Centre for Advanced Mediterranean Agronomic Studies (CIHEAM), Av. de Montañana, 1005, 50059 Zaragoza, Spain; 3Observatorio Para Una Cultura del Territorio, Calle Duque Fernán Nuñez, 2-1, 28922 Madrid, Spain; vero.hj@observatorioculturayterritorio.org; 4School of Agronomy, Food and Biosystems (ETSIAAB) at the Polytechnic University of Madrid (UPM), Campus Ciudad Universitaria Av. Puerta de Hierro, nº 2–4, 28040 Madrid, Spain; 5Instituto Andaluz de Investigación y Formación Agraria, Pesquera, Alimentaria y de la Producción Ecológica (IFAPA), Cam. de Purchil, s/n, 18004 Granada, Spain; samir.sayadi@juntadeandalucia.es

**Keywords:** public universities, sustainable food systems, young consumers’ perceptions, green public procurement, principal component analysis, cluster analysis

## Abstract

The agri-food system needs to transition into a more balanced system that takes into account economic, social, and environmental factors. Young people are a key demographic group to consider as they are open to new trends of consumption, including sustainable buying practices. Public universities can play a significant role in promoting sustainable and healthy eating habits among students. In this paper, we focus on the perceptions of young people regarding sustainable food in the Madrid Region. We conducted a survey using a questionnaire-based approach among 1940 students in 2022. The results highlight that young consumers are highly concerned about food sustainability. They perceive sustainability as local and non-processed foods. However, this perception varies among young consumers, and we identified five different consumer profiles. Principal component analysis and cluster analysis provide insights into potential actions that universities can take to promote sustainable and healthy eating habits among students.

## 1. Introduction

The sustainability of the food system is one of the main challenges facing humanity [1,2]. The agri-food system needs to transition toward a more balanced system from an economic, social, and environmental point of view [3,4]. The Green Deal 2019–2024 [5] and the Farm to Fork strategy [6] address this challenge at the European level.

The perceptions of the population about the existing food systems and their willingness to change their food habits are key in this transition. Fuchs and Lorek (2005) [7] stated that sustainability is about sustainable consumption rather than sustainable production. However, consumption can compromise sustainability [8].

Achieving changes in individuals’ consumption patterns and reductions in the volumes of consumption have been referred to as “strong sustainable consumption” or “deep green consumerism”. Consumers not only seek to purchase eco-friendly alternatives but, more importantly, they seek to reduce their overall level of consumption [9,10]. On the other hand, “weak sustainable consumption” or “shallow green consumerism” only entails an improvement in consumption efficiency and does not necessarily change their overall level of consumption [9,10].

Young people are a special demographic group to consider in the transition toward sustainable food consumption (SFC) [11,12], and public universities are an environment in which people engage with the food system and make decisions about consuming food. Some studies have focused on young consumers and food habits transition [13,14,15]. However, more information is necessary to understand this key population and to support their transition toward more sustainable food habits. The perceptions and contexts of these young consumers are different in different regions; this study aims to fill information gaps in the relevant literature regarding young people from Spain.

Additionally, public universities are demanding information to plan new sustainable food services. This paper focuses on filling information gaps to support the role of green public procurement and public universities in the transition toward a sustainable food system (SFS). According to FAO [16], an SFS is profitable throughout (economic sustainability), has broad benefits for society (social sustainability), and has a positive or neutral impact on the natural environment (environmental sustainability).

This paper aims to identify and characterize the perceptions of young people aged 18 to 35 [17] and in university to support strategies to promote the transition to a more sustainable and healthy food model.

Spanish public universities are committed to the notion of sustainability [18], but ensuring the sustainability of their food services is a big challenge for them. This paper focuses on the region of Madrid (RM), the region with the largest number of university students in Spain (17%) [19]. The research questions we sought to find answers for are listed below:

RQ1. What are the perceptions of university students on food sustainability?

RQ2. What eating habits would they be willing to change?

RQ3. What kind of profiles can be identified among young university students about sustainable food systems?

## 2. Background

### 2.1. Healthy and Sustainable Diets

Sustainability is threatened by some dietary patterns that include more calories, animal-sourced foods, and ultra-processed foods, and these dietary patterns result in high environmental footprints and high levels of food waste [20,21,22]. Food choices are linked with human health and the health of our environment [23,24,25]. Numerous studies have highlighted the role of consumers in promoting SFC [25,26,27,28].

The World Health Organization (WHO) defines healthy and sustainable diets as “those that are high in vegetables, fruit, and whole grains, with limited intake of saturated fat, trans fats, sugar, and salt” [29]. There is an increased recognition that systemic changes are required to improve access to healthy, sustainable diets [25,30,31,32]. The European Commission believes that enabling consumers to play an active role in making the EU food system sustainable is crucial [1].

Food habits are hard to change [33], and the food environment is a key driver of food choices [26]. The WHO encourages the creation of a healthy food environment through policies and investment plans that include encouraging consumer demand for healthy foods and meals and promoting appropriate infant and young child feeding practices [29].

In surveys and empirical studies, young people have emerged as one of the demographics most interested in adopting changes to a more sustainable and healthy diet [34,35,36,37,38,39]. Emerging evidence suggests a link between young people’s interest in alternative food production practices (e.g., growing foods locally using sustainable agricultural techniques) and dietary quality [22,40].

### 2.2. Sustainable Food Systems in the University

Education is the most promising mechanism for cultivating a more sustainable future [41,42]. Knowledge and motivation are essential for promoting pro-sustainability actions [43]. Therefore, increasing students’ levels of sustainability knowledge should be a top priority for institutions of higher learning [44]. Many universities are working on integrating sustainable development into their institutions, university life, and the curriculum of undergraduate and postgraduate degrees [45,46]. In the short term, higher education is recognized as a key vehicle for achieving sustainability [47]. Food services are sometimes excluded because they are considered individual choices. However, new restaurant policies can create positive impacts on the offered restaurant services and push for more sustainable practices [48].

The period of life that students spend in university, usually between the ages of 18 and 24, is characterized by major changes in lifestyle. Students take responsibility for their nutrition for the first time, become more independent, separate from family members, and change their social circle, which may influence them to change their established habits and adopt a new lifestyle [49,50].

Universities can offer a food environment that promotes healthy eating habits among students, such as interventions that promote healthy eating behaviors and improve access to healthy foods in the university environment, thereby facilitating the construction of eating patterns that will follow throughout adulthood. Therefore, universities should provide students with skills, knowledge, and encouragement to address sustainability challenges in a globally complex world [42,51,52].

Additionally, according to the European Green Deal [5], public universities can contribute to SFC through actions to promote green public procurement (GPP) policies that encourage the development of healthy and sustainable food environments [53,54]. Public authorities are major consumers in the European Union. By using their purchasing power, they can make an important contribution to sustainable consumption and production. At the European level, GPPs are regulated by Directive 2014/24/EU [55] on public procurement. This regulation seeks to ensure greater inclusion in the procurement process of issues like environmental protection, combating climate change, public health, and other social and environmental considerations.

In 2019, the Government of Spain approved the Green Public Procurement National Plan (2018–2025) [56]. This plan responds to the need to incorporate ecological criteria in public procurement. It includes twenty areas of action; the first one is food and catering services. More specifically, Spanish universities have been committed to sustainable development since 2004. This year, the Conference of Rectors of Spanish Universities, the most important association of Spanish Universities, launched a sustainable development working group. In this sense, more than 85% of Spanish universities have specific staff working on these topics and 98% have specific budgets for sustainable development [57]. Spanish universities tend to include environmental criteria in the public procurement contract specifications and regularly organize awareness and media campaigns [58]. More than 21% of Spanish universities have differing initiatives in place related to environmentally responsible procurement [59].

Food services on university campuses are particularly interesting spaces in which to work on sustainability, for several reasons. Firstly, the catering sector is a significant contributor to greenhouse gas emissions [60]. A recent study of food in public school canteens in Italy found that most of the environmental impacts of canteen meals derive from the presence of animal-based food and the use of energy for food preparation. This study also found that up to 1/3 of the prepared meal was wasted, especially vegetables [61]. Secondly, canteens are essential services for the functioning of the campus [60]. Canteens are integrated into academic life and many students, researchers, teachers, and administrative staff rely on canteen services, especially in isolated or non-urban campuses. Moreover, habits established during the university period can be tracked forward into later life [62]. Finally, educated people are more inclined to make sustainable food choices and adopt a sustainable diet [63].

GPP is one of the most powerful instruments of universities to foster this transition toward greater SFC. University canteens are not just spaces for eating. They are connections between SFS and academic disciplines, campus operations, and community partners [64,65]. Mikelsen et al. (2017) [66] suggest that organic conversion of public canteens may be a good opportunity to promote healthier eating in public catering. Universities can offer an enabling environment for more sustainable consumption.

### 2.3. Youth and Sustainable Food Habits

According to the International Day of Youth, “Transforming Food Systems: Youth Innovation for Human and Planetary Health”, the United Nations considers that youth (until 35 years old) engagement is key to the transformation of food systems [67]. Young people have growing importance and decision-making power in today’s households, and they respond to the changing environment, globalization, and their impacts on consumption [12,68,69]. Consumption patterns that affect one’s health and that of the planet are instilled early in life [67]. Youth are a key demographic group to consider as they are open to moving towards a new way of life including sustainable buying practices, reducing waste, and changing their dietary behavior [68,70,71].

The sustainable practices of young people rely on their perceptions and knowledge of sustainability [72,73]. Siegrist and Hartmann (2019) [27] conclude that increasing consumer knowledge about the environmental impact of foods may lead to more SFC. Young generations are increasingly conscious in terms of the ethical, environmental, and social impacts associated with their diets, although they may not fully understand which behaviors offer the largest environmental benefit [2,11,14]. However, some other papers highlight that there is a lack of interest in the younger generations for food habits and dietary changes [74]. The decision-making process depends on the consumer’s social responsibility (extrinsic) in addition to individual needs like taste, price, convenience, and health (intrinsic) [14,75].

The bibliography shows the heterogeneity of young people about SFC. Savelli et al. (2019) [76] identify four clusters of young consumers, namely healthy and certified food consumers, comfortable consumers, saver consumers, and innovative consumers. Bollani et al. (2019) [77] identify four groups of millennials (a population cohort born between 1981 and 1996) with specific perceptions about food sustainability: "Socio-Nature Sensitives”, characterized by a high level of attention to the socio-economic dimension and sustainable ways of food production; "Info-Supporter", very sensitive to labeling and warranty systems; "Proactive-Oriented", interested in innovative activities; and "Indifferent Millennials", assigning the issue, in general, a low level of importance.

The implementation of more sustainable food services on university campuses depends on the perceptions, uses, and demands of the users of food services. A manifestation of concern for the environment can be eliminating some products from consumption or giving up the purchase of environmentally unfriendly food products [78].

SFC has been linked to different consumption dimensions, such as consumption of local products, decrease in meat and processed product consumption, increase in fruit and vegetable consumption, and general consumption of products that have small ecological, carbon, and water footprints [62,79].

Meat consumption and animal-based protein are some of the most cited issues in the literature on sustainable food challenges. Younger people were more likely to perceive the recommendations to limit red and processed meat to have a high environmental benefit [2], although this did not correspond to being more ready to adopt these behaviors. The interest in plant-based diets amongst younger consumers may be explained by the emergence of nutrition and dietary trends within urban areas, and concern about climate change or animal welfare [68].

The use of plastic in food packaging is another relevant issue in the reflections on sustainability, since packaging is such a central and visible aspect of human consumption [80,81]. Consumers have an increasing environmental awareness and a positive willingness to pay for packaging alternatives [82]. Wierzbinski et al. (2021) [78] highlight that the popularity of some pro-ecological behaviors increases, i.e., buying products with less plastic packaging, limiting water consumption, and limiting wastage of water and food as well as waste segregation. Boesen (2019) [83] highlights that consumers focus on packaging rather than consumption or transport.

Reducing food waste is seen as an important way to lower production costs, increase food system efficiency, improve food and nutrition security, and contribute to environmental sustainability, and several studies have previously shown an unacceptable plate waste index in university canteens [50,84].

Finally, the consumption of local products and reducing the number of kilometers transported for food is another key issue about the sustainability of the food chain. Young adults had a positive attitude to local and traditional food products [85,86]. Kovács et al. (2022) [14] identify a segment of young consumers that has a positive view of local products and has a preference for buying local products.

In this transition toward a sustainable lifestyle, there are some barriers: (1) the scarcity of sustainable options and the affordability and convenience options, (2) the lack of awareness and knowledge among peers and the public about sustainability and its importance, (3) the difficulty in distinguishing between products, (4) cost of sustainable food, [63,70,87,88]. Additionally, changing the habits of consumption is hindered by the routine of buying and past experiences [89]. On the other hand, the more we consume ecologically and sustainably, the more we buy out of social commitment [90].

Beyond the individual approach, public policies can develop important strategies promoting healthy and sustainable eating and food consumption habits in young people [91,92,93]. Ruben et al. (2021) [94] and Trewern et al. (2021) [95] highlight strategies toward food system transformation: raise ambitions, harmonize goals and improve connectivity, strengthen responsiveness from linear agri-food value chains to circular food systems and anchoring governance, subsidize regenerative and agroecological production, incentivize consumers, and provide healthy and sustainable food in public canteens (schools, universities, etc.).

### 2.4. Theoretical Framework

To address the gap in sustainable food services in public institutions, the theoretical framework deals with intrinsic and extrinsic drivers [14,75]. Firstly, the intrinsic drivers are focused on the three attitude components: cognition, affect, and behavior [96]. Cognition is considered in terms of the perception of food sustainability; affect in terms of the concern about the sustainability of food habits; and behavior in terms of willingness to change food habits.

Secondly, the extrinsic drivers are represented by the public universities. The university food services and the commitment of public universities to sustainability establish the extrinsic context to complete the analysis.

According to this background, the starting hypotheses to deal with the RQ are:

**H1**.*University students are concerned about the sustainability of the food system* [2,11,14,97].

**H2**.*The main habits they would like to change include “strong sustainable consumption” and “weak sustainable consumption”* [9,10].

**H3**.*Several profiles are willing to change. Just a small group wants to “shop as usual” or is indifferent* [75].

Lastly, based on the gathered results, the present paper includes some implications and recommendations to support the transition toward an SFS from the public universities.

## 3. Methods

According to the aim of this research, the targeted population of this analysis was universities and young people. Young people have been considered from 18 to 35 years old in the United Nations states by the International Day of Youth. The fieldwork was performed in the region of Madrid. It has the largest number of university students in Spain—almost 279,000 [19]. Finally, the area of study is focused on public universities, as one of the points of this paper is the GPP and the role of public actions in fostering SFC.

To identify and characterize the perceptions of young people in the university context, a survey was applied to 1940 students via a questionnaire-based approach according to the convenience sampling procedure. It was implemented in four universities: Universidad Complutense de Madrid (24%), Universidad Rey Juan Carlos (36%), Universidad Autónoma (24%) and Universidad Politécnica de Madrid (16%), from April to June 2022. In total, 90% of the surveyed people were younger than 24 years old. The fieldwork was performed by the local government of Madrid under the umbrella of the actions of the Food Wave project. Food Wave is an initiative co-funded by the EU and coordinated by the Municipality of Milan (Italy). The Food Wave consortium involves 18 cities and municipal agencies across 17 countries. The aim was to foster the transition toward a more SFC and increase awareness of sustainable food consumption and production. After a pilot test of the questionnaire, the final version, with dichotomous (0–1) closed questions, addressed five main issues: sustainability conceptualization, food habits, main criteria for food selection, level of concern about food habits, and willingness to change their habits. During the 12 workshops held on university campuses, Food Wave project technicians carried out face-to-face surveys.

Based on the data gathered, a descriptive, regressions, principal component, and cluster analysis were performed with IBM SPSS (v.26). Principal component analysis was used to distill data into principal components, and chosen for its ability to preserve variance in reduced dimensions and illuminate underlying patterns in sustainability conceptualization and food habits. The selected components were with eigenvalues greater than one. A varimax rotation with Kaiser normalization was run previously for the data interpretation. The principal component analysis was focused on the relationships between food sustainability conceptualization, main criteria for food selection, and willingness to change younger their habits. The individual score of each component was transformed on a scale of 0 to 9 before calculating the average and representing the distribution [96]. Finally, a cluster analysis targeting the extracted variable factors with eigenvalues greater than one with a K-means optimization was applied to group the young respondents in clusters according to the selected factor variables, justifying its use due to its superiority in revealing patterns and segmentation in this context. Afterward, an ANOVA test reinforcing the choice of these factors was carried out. In addition, and to group the young respondents according to their similarities, three cluster profiles were created. All these analyses are relevant to support strategies to leap toward healthy and sustainable diets in the university.

## 4. Results

### 4.1. RQ1. What Is the Perception of Food Sustainability of University Students?

Seventy-six percent of respondents think about the sustainability of the food they consume, twenty-eight percent are very concerned about the sustainability of their food, and six percent never think about it. When obtaining information about food sustainability, we found that 35% of respondents follow food profiles or information on social media.

The conceptualization of sustainable food has an interpretation that implies, almost equally, the consumption of local food (62%), buying locally (59%), and consuming less processed products (59%). A lower consumption of animal protein is only considered by 28% of respondents.

Taking into account these conceptualizations of what sustainable food implies, 68% self-perceived their practices as somewhat sustainable, while 22% perceived their practices as not very sustainable. Nine percent consider their practices to be very sustainable.

Self-perception of sustainable consumption habits is related to conceptualizations of sustainability consumption like shopping locally and consuming less meat (F = 3.155; Sig. = 0.014). Those who perceive their consumption habits as less sustainable do not give as much importance to meat consumption in the conceptualizations of sustainability but focus more on local consumption (F = 2.529; Sig. = 0.039).

### 4.2. RQ2. What Eating Habits Would Be Willing to Change?

In general, price (83%), quality (74%), and, to a lesser extent, taste (54%) are the most important criteria for young participants when making purchasing decisions. The traditional foods were not found to be highly relevant in the preference criteria of the surveyed consumers.

The analysis also reveals that those who consider that they have more sustainable practices stand out because they do not give as much importance to economic issues and pay more attention to the proximity of the products (F = 14.073; Sig. = 0.000). On the other hand, those who consider themselves to have somewhat sustainable practices focus on quality, provenance, and value practicality less (F = 10.514; Sig. = 0.000). Those with a less sustainable self-perception focus their purchasing criteria on practicality. They do not attach importance to quality or origin (F = 19.569; Sig. = 0.000).

About 54% of participants reported a willingness to change more than four habits to reach a more healthy and sustainable diet. The main changes people are willing to make are as follows: selecting food with less plastic packaging (78%), eating less processed food (69%), buying food from local shops (62%), opting for seasonal food (60%), reducing meat consumption (42%), selecting food with an early expiry date (38%) and being willing to pay more for more sustainable products (24%) (Figure 1).

Individuals who exhibit higher levels of concern about the sustainability of the food system are more likely to report reducing their meat consumption, pay more for more sustainable food, and buy seasonal food (F = 18.357; Sig. = 0.000). However, they would not reduce the consumption of processed foods.

Those who are somewhat less concerned about sustainability prefer to change their habits toward less processed product consumption, but they would not consume less meat, and they would not pay more for more sustainable food (F = 9.843; Sig. = 0.000).

### 4.3. RQ3. What Kind of Profiles Can Be Identified among Young University Students about Sustainable Food Systems

Principal component analysis (Kaiser Meye Olkin = 0.61; Sig. = 0.000) (Table 1 and Table 2) of the variables studied relating to the conceptualization of sustainability and purchasing habits shows five components that can be associated with as many profiles.

The first component could be associated with “Local consumer habits”. For this component, sustainability entails local products. These consumers would be willing to include more seasonal products. The mean of these values is 4.6 on a scale of 0 to 9 (Figure 2).

The second component, “Less animal protein”, puts more emphasis on reducing animal protein (Figure 3). Thus, the mean value of this profile is 3.4 on a scale of 0 to 9.

The third component, “Quality”, is characterized by an increased concern for food quality. These consumers pay attention to the provenience and the food quality. Convenience is not a priority for them. They would be willing to pay more for sustainable products. The average of the scores of this cluster is 4.8 (Figure 4).

The fourth component, “Non-processed food”, has the highest average score. This component is focused on non-processed foods. The average score is 5 on a scale of 0 to 9 (Figure 5).

Finally, the fifth factor focuses on the purchase of food according to what has been carried out in the past, consumption as usual (“Traditional”). People buy certain products out of habit, without highlighting other sustainability considerations. This is the factor with the least weight among the surveyed population (1.9 average scores) (Figure 6).

To complement these results, a cluster analysis was performed on the factorial scores. Three main clusters arose from the data. The first (*n* = 749), labeled “Ecohealthy”, is mainly focused on vegetarian and non-processed food. The second cluster (“Loquality”) is characterized by their interests in local, seasonal, and quality products (*n* = 949). Finally, the last cluster includes those young people who try to buy food as usual. They are not concerned about their food habits. It is the smallest cluster (*n* = 242) (Table 3, Table 4 and Table 5).

## 5. Discussion

Eating habits are key drivers in the transition toward a more SFS [7,8,23]. Section 2 highlights some foods associated with the FAO definition of SFS [16]. Local, seasonal, and less processed food can be associated with social and economic benefits for society [23,24,25]. Reducing animal protein consumption also has a positive or neutral impact on the natural environment [20,21,22].

In terms of perception (RQ1), surveyed young university students are committed (H1) to the sustainability of their food habits, as in Culliford and Bradbury (2020) [2], Pflugh et al. (2019) [11], Kovács et al. (2022) [14] and Kowalska et al. (2021) [15]. Only 6% of the surveyed population never ask themselves this question, as the “Indifferent” profile in Bollani et al. (2019) [77]. It is interesting to note what they understand about sustainability. Local production and consumption and the degree of food processing are central to their consideration of sustainability [72,73]. In this sense, as in the European Farm to Fork strategy, short food supply chains are potential options for them [22,40]. The first component of PCA highlights this idea.

According to the intrinsic motivations [14,75], young people are in a good position to support the transition to a more sustainable food system (RQ2). The consumption of ultra-processed food and animal protein are some of the lines that appear increasingly recurrently in scientific studies about unsustainable food habits [20,21,22], and some young people (69%) are willing to change their habits about the first line. Furthermore, the fourth component (“Non-processed food”) in our PCA has the highest average—5 points. However, regarding animal protein, just 42% would change their practice (also, the vegetarian component of PCA has a low average of 3.4).

Following H2, cluster analysis presents a combination of sustainable and healthy diet approaches in the two biggest groups—cluster 1 and cluster 2. The first one with vegetarian and non-processed food entails “strong sustainable consumption” or “deep green consumerism” [9,10]. Cluster 2 involves relevant points (local, seasonal, and quality products) that are closer to “weak sustainable consumption” [9,10].

The cluster analysis identifies five profiles (RQ3) “Local consumer habits”, “Less animal protein”, “Quality”, “Non-processed food” and “Traditional”. In terms of changing consumption habits, there are some important challenges. However, there is room for optimism in the component named “Traditional” (comfortable consumers in Savelli et al., 2019 [76]). This is cluster 3, and it represents 12% of surveyed young people (H3). The low score in this component leaves open the possibility of working on changing habits and incorporating new practices and forms of consumption [68,69,70,71], contrary to what is stated in Vanhonacker et al. (2013) [74].

Extrinsic conditions [14,75] are key in this transition, and the university is a key actor. This analysis opens up two lines of action to be worked on from the university sphere as a space for promoting new habits. On the one hand, focusing on food quality in the canteens, and on the other, including healthy food in the range of university food services.

Slow food [98], flexitarian and vegetarian [99], or anticonsumption [39] are some diet alternatives to support the transition. The flexitarian abstains from eating meat occasionally without abandoning it. Vegetarians, on the other hand, follow a strict meat-free diet, and vegans abstain from consuming any animal-based products. Finally, as overconsumption has negative effects on ecological balance, social equality, and individual well-being, reducing consumption levels among the materially affluent is an emerging strategy for sustainable development. Today’s youth form a crucial target group for new types of food systems.

The more people consume ecologically and sustainably, the more they buy out of social commitment [91]. Consequently, universities can break the doubts mentioned in Glover and Sumberg (2020) [69] about the temporal concern of young people, because when they are adults, they tend to become more traditional in their consumption habits instead of adopting new ones.

## 6. Implications and Recommendations for Public Universities

Universities can meet the expressed demands of consumer profiles that are interested in a diet with a lower animal protein content (vegetarians and flexi-vegetarians), as well as local and seasonal foods. Likewise, they should continue to focus on the reduction in single-use plastics in the canteen and cafeteria services. To this, GPP is a core tool [54].

On the other hand, universities have an important role to play in the process of raising awareness and transitioning consumption habits. They should focus on healthy food environments as defined by the WHO (2021) [100]. The results show the need for further work in consumer awareness and information. Some experiences have been already tested in situ at the universities under the framework of Feeding the Campus Operational Group [101], which promotes sustainable meals, farmer markets, and expertise workshops for public procurement. Aware consumers are in a more favorable position to make sustainable consumption choices. If they are unaware of the impacts of the different modes of production, and of the effects that choosing one product or another may have on the market, the most immediate and practical criteria will ultimately take precedence. In the same way that advertising encourages certain consumer habits by launching messages of success, enjoyment, or happiness associated with the purchase of certain products, it is not easy to find equally attractive and mobilizing messages for the promotion of more sustainable products. Approaching local and seasonal food consumption, increasing its presence in mass catering, and informing about it could contribute to strengthening consumer arguments in favor of more sustainable consumption [102].

## 7. Limitations of the Study and Identification of Potential Areas for Future Research in This Field

This research is focused on young students at university; future complementary research with a more representative sample and more age target groups can address other profiles coexisting on the campuses, like teachers, researchers, and administrative staff, as university food services address all their interests. Additionally, public policies are committed to green public procurement and food purchase is a key issue with relevant social, economic, and environmental impact. In this sense, more research is necessary to support this process.

## 8. Conclusions

A significant proportion of young people are committed to the sustainability of their food consumption habits (RQ1). This paper highlights that some young people are demanding deep and soft sustainability measures (RQ2). Just a small group does not want to change their food habits (RQ3).

Public universities have a key role in working with intrinsic and extrinsic drivers to promote food transition among young people. University public procurement can be an important catalyst for this transition towards sustainable food consumption offering more sustainable alternatives in their food services. Furthermore, if public universities implement sustainable food procurement, they would be able to influence the food market. Farmers and food producers would opt for new modes of production and even favor economies of scale that result in more accessible prices.

Additionally, as the challenge of food sustainability is to reach wider groups of the population, public universities can contribute to raising visibility, awareness, and knowledge about more sustainable food habits among young people on their campuses.

## Figures and Tables

**Figure 1 foods-12-04103-f001:**
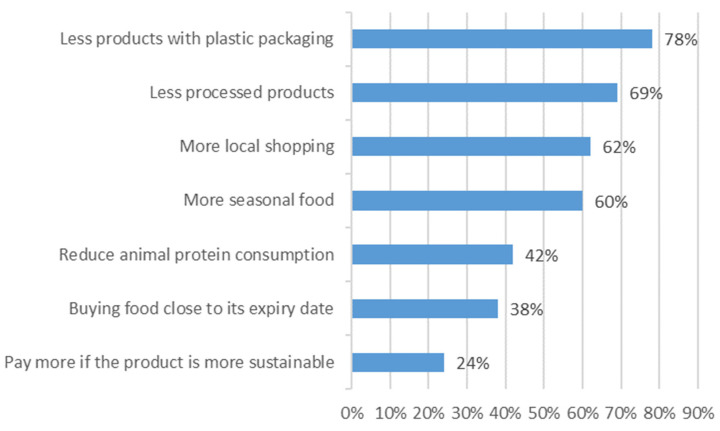
Percentage of young people are willing to change different habits regarding food (n = 1940). Values are ordered from the highest percentage to the lowest.

**Figure 2 foods-12-04103-f002:**
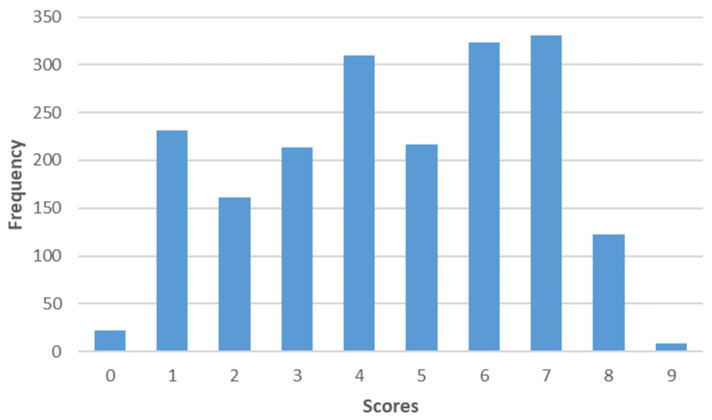
Distribution of the factor scores "Local consumer habits" on a scale of 0 to 9.

**Figure 3 foods-12-04103-f003:**
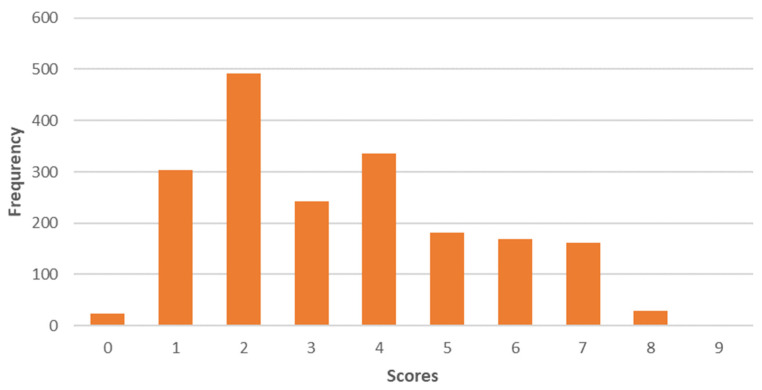
Distribution of the factor scores "Less animal protein" on a scale of 0 to 9.

**Figure 4 foods-12-04103-f004:**
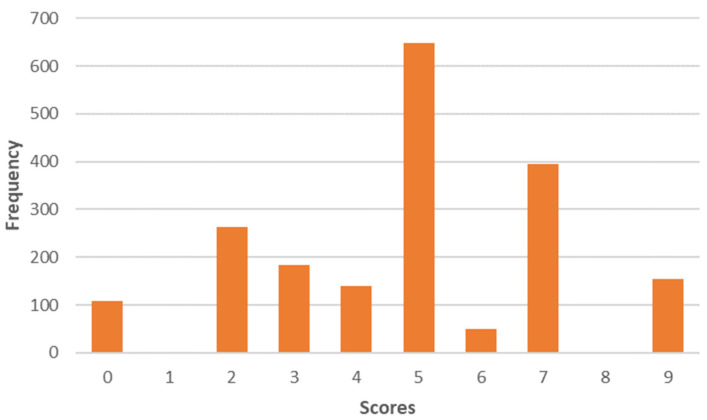
Distribution of the factor scores “Quality” on a scale of 0 to 9.

**Figure 5 foods-12-04103-f005:**
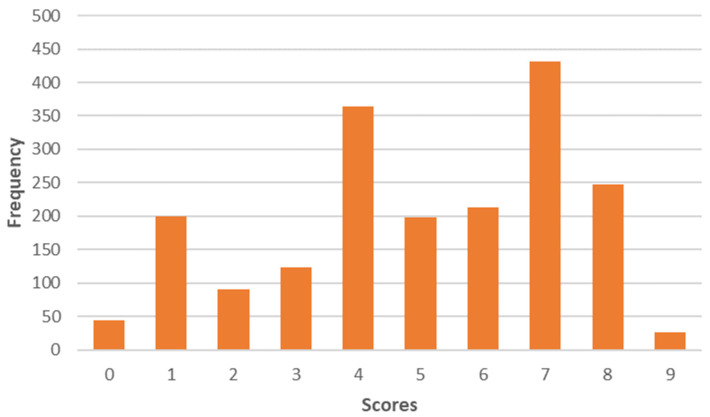
Distribution of the factor scores “Non-processed food” on a scale of 0 to 9.

**Figure 6 foods-12-04103-f006:**
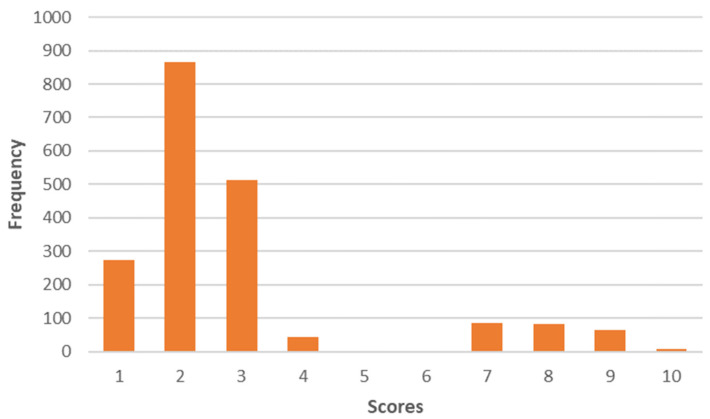
Distribution of the factor scores “Traditional” on a scale of 0 to 9.

**Table 1 foods-12-04103-t001:** Principal component analysis. This table presents the total variance explained. The five selected components have an eigenvalue higher than 1. The explained variance is 59.385%.

Component	Initial Eigenvalues			Rotation Sums of Squared Loadings		
	Total	% de Variance	Cumulative %	Total	% de Variance	Cumulative %
1	1.912	17.381	17.381	1.464	13.309	13.309
2	1.356	12.323	29.703	1.421	12.915	26.224
3	1.178	10.712	40.416	1.319	11.989	38.213
4	1.071	9.736	50.151	1.297	11.794	50.007
5	1.016	9.234	59.385	1.032	9.378	59.385
6	0.869	7.897	67.282			
7	0.816	7.414	74.696			
8	0.808	7.343	82.039			
9	0.731	6.643	88.682			
10	0.678	6.166	94.847			
11	0.567	5.153	100.000			

**Table 2 foods-12-04103-t002:** PCA. Rotated component matrix. This table shows the variables included in each component. It presents just the values higher than ±0.4.

	Components				
	1	2	3	4	5
A sustainable diet means…					
…eating little animal protein		0.775			
…local foods	0.700				
…less processed food				0.777	
Criteria when buying food…					
…quality			0.638		
…convenience	0.407		−0.480		
…shop as usual					0.948
…provenience			0.542		
What habit/s would you be willing to change to make your food more sustainable?….					
…reduce animal protein consumption		0.743			
… more seasonal food	0.782				
… less processed products				0.718	
…pay more if the product is more sustainable			0.600		

**Table 3 foods-12-04103-t003:** Cluster analysis. Final cluster centers. This table classifies into three clusters the respondents according to their factorial scores.

	Cluster		
	“Ecohealthy”	“Loquality”	Traditional
Local consumption	−0.52884	0.39357	0.09339
Vegetarian component	0.52774	−0.37842	−0.14941
Quality component	−0.47067	0.37199	−0.00201
Non-processed food component	0.30005	−0.24781	0.04310
Traditional component	−0.31237	−0.39352	2.50998

**Table 4 foods-12-04103-t004:** Cluster analysis. ANOVA. This table shows that all the variables have a significant relationship with the clusters.

	Cluster		Error		F	Sig.
	Sum of Squares	df	Mean Square	df		
Local consumption	179.294	2	0.816	1937	219.749	0.000
Vegetarian component	174.950	2	0.820	1937	213.252	0.000
Quality component	148.624	2	0.848	1937	175.353	0.000
Non-processed food component	63.080	2	0.936	1937	67.400	0.000
Traditional component	872.320	2	0.100	1937	8693.542	0.000

**Table 5 foods-12-04103-t005:** Cluster analysis. Number of cases in each cluster. This table quantifies the number of cases in each cluster.

**Cluster**	**“Ecohealthy”**	749
	**“Loqualitiy”**	949
	**Traditional**	242
Valid		1940

## Data Availability

The data presented in this study are available on request from the corresponding author.

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
