# Peer review of "The Gap in Sustainable Food Services in Public Institutions: The Perceptions of Young Consumers from Public Universities in the Madrid Region (Spain)"

_foods, 2023, doi:10.3390/foods12224103_

Round 1
Reviewer 1 Report
Comments and Suggestions for Authors
Dear authors, the paper is extremely interesting, especially as the concept of sustainable consumption is increasingly promoted nowadays, and young people are the ones who can become promoters of this way of consumption.
In the introductory part, I would suggest that the authors present the gap identified in the literature, especially as there have been studies, including in Europe, on this topic.
Pocol, C. B., Marinescu, V., Amuza, A., Cadar, R. L., & Rodideal, A. A. (2020). Sustainable vs. Unsustainable Food Consumption Behaviour: A Study among Students from Romania, Bulgaria and Moldova. Sustainability, 12(11), 4699.
I suggest that you refer to these studies that already exist in the literature and show how you have identified a gap, what the research issue is and how your study has led to the advancement of the literature.
In terms of methodology, the convenience sampling procedure has certain limitations because the sample is not representative in this case. Please provide arguments for the choice of this sampling method.
Also on the methodology side, the authors should present step by step the main steps of the cluster analysis. These are not explained at all from a theoretical point of view. They are only found in the results, which is not enough.
Please note this text is in brackets in some sections of the article:4Er-340 ror! Reference source not found.
Also, each cluster should have a name/label, depending on the characteristics of the reservoirs within the cluster.
Conclusions should be rewritten. In their current format, they are too general and do not reflect the results of the research obtained. The authors should also point out the limitations of the research, especially considering the sampling method used and the theoretical, practical and managerial implications of the study.
Reviewer 2 Report
Comments and Suggestions for Authors
Authors
The manuscript is a bit too long and there are parts that are redundant. A particular example of this appears in the Results section, where the description in the text is precisely the same as that shown in the figures.
There are no major editing problems. However, there are some small faux pas, which I will list below:
Abstract:
Line 22: Not 1,940 surveys. It should be 1 survey focusing/applied to 1,940 students via a questionnaire-based approach.
Introduction
Ls 47-53: This is the objective of the paper and should be in a separate paragraph.
Ls 54-60: This outlines the structure of the manuscript. It should be either not included (due to redundancy) or eventually merged to the objective (as per above).
Background
L 214: Please amend to Ruben et al. (2021)
Methods
L 228: Please amend the abstract based on the following “… a survey was conducted on a large group (n= 1,940), according…”
Results
L 290: “The 54% of participants reported…” This is not the most correct sentence. Better to change: Not "The" but "About"…
Ls 291-5: Not very important but pay attention to the following: Slight differences between text and figure.
Some error messages appear: Ls 327, 340-1, 348, 358-9
Discussion
Ls 387, 389, 406-7, 408, and may be more: Please cite numbers in ascending order.
L 449: “…a key issue with a lot of relevance…”. "a lot" should be replaced by a more formal/scientific term.
In general:
In terms of graphic presentation and tables, the figures are a little poor and should be more elaborate and clarifying.
Comments on the Quality of English LanguageApart from few bits, the level is reasonably good.
Reviewer 3 Report
Comments and Suggestions for Authors
We appreciate your submission of your research work to Foods. While the authors have gathered extensive data, it is important to note that the paper lacks a well-structured scientific framework. To enhance the manuscript, we kindly request that you consider the following suggestions:
1. Ensure that the research is centered around clearly defined research questions, and align the data analysis and discussion with these questions.
2. Establish a strong theoretical foundation for your study, as it currently lacks adequate theoretical support.
3. Develop hypotheses to empirically test the quantitative data collected in your study.
4. Revise the methodology section to provide a clearer explanation of how the research instrument was developed and how its validity and reliability were established. Additionally, justify the choice of statistical methods such as PCA and Cluster analysis over other available techniques like regression and inferential statistics.
5. Address the discrepancy between the high level of concern for sustainable food presented in your abstract (page 22) and the finding on line 255, which indicates that 76% only occasionally think about food sustainability.
6. In the discussion section, elaborate on how certain eating habits or changes in eating habits can contribute to economic, social, and environmental sustainability.
7. Create a separate section to discuss the implications and recommendations derived from your research after the discussion section.
8. Establish a dedicated section to thoroughly examine the limitations of the study and identify potential areas for future research in this field.
Round 2
Reviewer 3 Report
Comments and Suggestions for Authors
The authors have made significant changes to the manuscript, however some errors still exit and need to be addressed before it can be recommended for publication.
Research Questions and Hypotheses are not the components of methodology section. Research questions to be placed within Introduction section.
Hypotheses should be placed in the theoretical framework section. Each hypothesis should be substantiated with ample literature.
Findings should be discussed vis-a-viz research questions and hypotheses
